# Cytomegalovirus Immunoglobulin G Levels and Subclinical Arterial Disease among People Living with HIV in Botswana: A Cross-Sectional Study

**DOI:** 10.3390/biomedicines12050935

**Published:** 2024-04-23

**Authors:** Thato Moshomo, Onkabetse Julia Molefe-Baikai, Kara Bennett, Tendani Gaolathe, Sikhulile Moyo, Simani Gaseitsewe, Terence Mohammed, Shahin Lockman, Mosepele Mosepele

**Affiliations:** 1Department of Internal Medicine, Faculty of Medicine, University of Botswana, Gaborone Private Bag 00713, Botswana; onkabetsebaikai@gmail.com (O.J.M.-B.); mosepele.mosepele@gmail.com (M.M.); 2Bennett Statistical Consulting, Inc., Ballston Lake, NY 12019, USA; 3Botswana-Harvard Health Partnership, Gaborone Private Bag BO 320, Botswana; 4Department of Immunology & Infectious Diseases, Harvard T. H. Chan School of Public Health, Boston, MA 02115, USA; 5Division of Infectious Diseases, Brigham & Women’s Hospital, Boston, MA 02115, USA

**Keywords:** cytomegalovirus, HIV, cardiovascular, monocyte activation, endothelial injury, Africa, antiretroviral therapy

## Abstract

Cytomegalovirus (CMV) has been linked with increased cardiovascular risk and monocyte activation in people living with HIV (PLWH). This cross-sectional study aimed to compare CMV immunoglobulin G (IgG) levels between combined antiretroviral therapy (cART)-treated PLWH versus ART-naïve PLWH and those without HIV, and to investigate their associations with biomarkers of endothelial injury and carotid atherosclerosis, in Gaborone, Botswana. All participants were between 30 and 50 years old. Carotid intimal media thickness (cIMT) and biomarkers of endothelial injury and monocyte activation were also assessed. The association between quantitative CMV IgG and cardiovascular disease risk was assessed in multivariate logistic regression analysis. The results showed that the mean CMV IgG level among ART-naïve participants was significantly higher than both the cART group and controls. However, CMV IgG levels did not differ significantly between the controls and cART groups. Among PLWH, CMV IgG levels were associated with ICAM-1 levels and cIMT. Increases in CMV IgG among ART-naïve participants were significantly associated with increases in log VCAM-1. In conclusion, CMV IgG levels are elevated among PLWH in sub-Saharan Africa, and higher levels are associated with biomarkers of endothelial injury and cIMT. Future research should investigate the long-term impact of elevated CMV IgG among PLWH.

## 1. Introduction

According to the World Health Organisation report (2023), at least 39 million people live with HIV globally [1], and this population is known to be at an increased risk of cardiovascular disease [2,3,4]. Traditional cardiovascular risk factors incompletely account for this increased risk [2,5,6,7,8,9]. Persistent immune activation and inflammation, driven by HIV reservoirs, microbial translocation, and viral coinfections like cytomegalovirus (CMV), declines but fails to normalize during antiretroviral therapy (ART)-mediated viral suppression and has emerged as a potential contributor to increased cardiovascular disease (CVD) risk [6,10,11,12]. CMV-HIV co-infection occurs in over 90% of people living with HIV (PLWH); however, the majority of patients are asymptomatic [13]. Although asymptomatic CMV infection appears to play a smaller role in CVD in the general population, multiple studies have linked CMV to CVD in PLWH [7,14,15], though the mechanisms remain unclear.

Recent meta-analyses have established links between CMV burden (assessed by seropositivity or levels of CMV-reactive antibodies) and CVD [7]. Hsue et al. [16] reported correlations between T-cell responses to CMV and increased carotid intimal media thickness (cIMT) in PLWH in the US, while Knudsen et al. [17] described a direct association between levels of CMV-reactive antibodies and cIMT in Danish patients stable on ART. Despite studies demonstrating associations between CMV immunoglobulin G (IgG) antibodies and subclinical carotid artery disease among PLWH, the precise relationship between CMV IgG levels, endothelial dysfunction, and arterial disease remains controversial. Furthermore, these associations have not been fully explored across different groups in populations with a high HIV burden.

Therefore, we performed a retrospective cross-sectional study that compared CMV IgG levels in a well-characterized cohort of adult PLWH and a matched group of HIV-uninfected individuals in Gaborone, Botswana [18,19]. Additionally, we investigated the associations between CMV IgG levels and cIMT, as well as biomarkers of endothelial injury/inflammation, including vascular cell adhesion molecule 1 (VCAM-1), intercellular adhesion molecule 1 (ICAM-1), e-selectin, and monocyte activation (soluble CD163 [sCD163]) among all participants, with adjustments made for the three study groups.

## 2. Materials and Methods

### 2.1. Study Participants

We measured cytomegalovirus (CMV) immunoglobulin (IgG) levels and biomarkers of endothelial injury and monocyte activation using samples obtained from participants in a previously described study evaluating sub-clinical carotid atherosclerosis and immune activation among adult PLWH compared to HIV-uninfected controls in Gaborone, Botswana [18,19]. All participants self-identified as black Africans, and it is well known that Botswana’s population is largely ethnically homogenous [20]. From these, we defined the following 3 study populations:(1)PLWH on fully suppressive combination ART (cART);(2)PLWH who are ART-naïve, and;(3)HIV-uninfected controls.

PLWH were recruited from the Princess Marina Hospital Infectious Disease Care Clinic (PMH-IDCC) in Gaborone, whilst HIV-uninfected controls were recruited at a voluntary HIV testing center located less than 1 km from PMH-IDCC between February 2014 and April 2015. Given our high uptake of ART nationwide and the attainment of the UNAIDS 95:95:95% targets [21], we generally expect a low number of participants to be ART-naïve in Botswana.

The inclusion criteria included participants aged between 30 and 50 years old and able to lie down for carotid ultrasound scanning. For PLWH on antiretroviral therapy (ART), the following was required: documentation of HIV diagnosis (either confirmed dual-positive ELISA or HIV-1 RNA viral load >400 copies/mL), on ART for at least one year, on the same cART regimen for at least 6 weeks, and viral load <400 copies/mL within 6 months prior to the study visit. For PLWH who were ART-naïve, a confirmed dual-positive ELISA result using kits from 2 different manufacturers was required for enrolment (as per the local standard of care). For the HIV-uninfected arm, documented HIV-negative status (dual-negative ELISA) was required (as per the local standard of care).

The exclusion criteria included were (within 7 days prior to enrolment) diarrheal illness, the use of antibiotics/immunosuppressives/probiotics, and inflammatory bowel disease, mainly because most of these participants were also screened in a gut microbiome sub-study (reported elsewhere). Pregnant women were also excluded (based on self-reports).

### 2.2. Study Procedures

This is a retrospective cross-sectional study embedded within a completed cross-sectional study [18,19] in which data such as blood pressure, body mass index, and plasma samples were collected during one visit. The main cross-sectional study provided information on participants’ medical history and CVD risk factors such as age, sex, smoking status, lipid panel, glycosylated hemoglobin, and associated treatments. Family history of stroke and myocardial infarction among both 1st- and 2nd-degree relatives was also already available. Consented and enrolled participants also previously underwent cIMT measurement; blood was drawn and frozen at −80 degrees Celsius. Thawed plasma from EDTA tubes was used to ascertain quantitative levels of CMV IgG by ELISA (GenWay Biotech CytomegaloviruS (CMV) IgG Enzyme immunoassaY test kit Catalog Number: 40-052-115031, San Diego, CA, USA) following the kit manufacturer’s instructions. VCAM-1, ICAM-1, and e-selectin levels were already available from the prior study [18,19] on the same participants.

### 2.3. Carotid Imaging

The mean common cIMT was used as a summary measure of the observed degree of atherosclerosis among HIV-infected participants [18,19]. The mean cIMT was ascertained at the distal 1 cm of the common carotid arteries on images obtained in B-mode along the lateral, anterior, and posterior longitudinal sections of the common carotid artery bilaterally (right and left common carotid) as per the 2008 American Society of Echocardiography Carotid Intima–Media Thickness Task Force protocol.27 A. 

A Sonosite M-turbo© ultrasound machine (FUJIFILM Sonosite Inc., Bothell, WA, USA) connected to an 8–12 MHz linear probe was used to obtain still images at the beginning of the R wave, and Sonocal© (version 5 of 2011) was used to measure cIMT in auto-mode as per the manufacturer’s instructions. Plaque was defined as any focal atherosclerotic lesion >1.5 mm on a still image obtained at the beginning of the R-wave.

### 2.4. Statistical Methods

Data analysis was performed using the Statistical Analysis Software (SAS˝), version 9.4 (SAS Institute, Cary, NC, USA). The code used is not available.

Power calculations for the main study have been presented previously [19]; however, no formal power calculations were performed in the current sub-study. The baseline characteristics of the three study groups (ART-experienced PLWH, ART-naïve PLWH, and controls) were summarized and compared. Categorical variables are presented as percentages (%) and compared using Fishers exact test, while continuous variables are presented as means (SD) for normally distributed variables or as medians (IQR) for skewed distributions. Continuous variables were compared using a two-sample *t*-test or non-parametric Wilcoxon rank sum test, based on the distribution of the data.

The distribution of CMV IgG levels across the three study groups was examined using side-by-side boxplots and assessed for normality and for outliers. A log transformation was applied to improve the fit to the normal distribution. Five known outliers in the CMV IgG data (all > 500) were excluded from the analyses based on communication with the laboratory on suspicion of a lab error affecting those five specimens.

Associations between CMV IgG and the following covariates of interest were individually assessed for each study group: age, sex, ART duration, number of ART switches, protease inhibitor (PI) versus non-nucleoside reverse transcriptase inhibitor (NNRTI)-containing regimen, nadir CD4, current CD4, waist–hip ratio, cigarette smoking, diagnosis of hypertension, and glycosylated hemoglobin (HbA1c). Correlations (Pearson or Spearman’s, as appropriate) assessed associations between continuous covariates, and a two-sample *t*-test or non-parametric Wilcoxon rank sum test, as appropriate, assessed categorical covariates in each of the study groups. The choice of parametric versus non-parametric tests was based on the assessment of the distribution of CMV IgG as described above.

The association of CMV IgG with arterial disease and monocyte activation was assessed among all participants and adjusted for each of the three study groups. Arterial disease was measured using markers of endothelial injury [ICAM-1, VCAM-1, or e-selectin] and the mean common carotid intima thickness (cIMT) of both the left and right carotid arteries, while monocyte activation was assessed using soluble CD163 (sCD163). The presence of plaque was only found in 4 participants across all study groups and thus was not included as a marker of arterial disease. Five separate univariable linear regression models assessed whether CMV was associated with each of these measures of arterial disease or monocyte activation. Model assumptions were checked, and the log transformation of relevant variables was considered. Multivariable models assessed for potential confounders, including age, sex, ART duration, number of ART switches, PI versus NNRTI-containing regimen, nadir CD4, current CD4, waist–hip ratio, cigarette smoking, diagnosis of hypertension, and HBA1C. Confounding variables were identified using the “change-in-estimate” approach (i.e., any variable that, when added to the model, causes the parameter estimate for CMV IgG to change by more than 10%). Potential outliers were identified, and sensitivity analyses assessed whether conclusions were affected by these outliers. The final multivariable models for each outcome included CMV IgG, study group, and any variables with a *p*-value < 0.15 or deemed to be confounders.

Finally, each of the final multivariable models was re-assessed among the ART-experienced group only and separately adjusted for ART duration, number of ART switches, type of ART regimen, and nadir CD4.

## 3. Results

### 3.1. Demographic and Clinical Characteristics of Participants

Among the original 472 participants in the source cross-sectional pilot study, 188 (39.8%) had CMV measured (simple random sampling), and 5 of these had CMV levels above 500 IgG. At the laboratory’s request, these were excluded from analyses presented in this report. Thus, 183 (38.7%) participants had a valid CMV measurement available and were included in analyses presented in this report.

Among the 183 CMV analysis participants (Table 1), 79 were HIV-uninfected controls (45.6% female, mean age 38.2 years), 68 were PLWH and on ART (51.5% were female, mean age 38.7 years), and 36 were PLWH who were ART-naïve (58.3% female, mean age 37.8 years). Despite an imbalance in the number of participants per group resulting from the availability of CMV immunoglobulin per HIV sub-group from the source study population groups, there was no statistically significant difference detected between the three groups.

PLWH on cART had more comorbidities (hypertension, chronic kidney disease, and dyslipidemia) than those in the other two groups. cIMT was not associated with any of the three HIV sub-groups (*p* = 0.059). Both PLWH who were ART-naïve and on c-ART had higher ICAM-1 and log VCAM levels than those who were HIV-uninfected. The rest of the characteristics are as shown in Table 1 below.

Continuous measures are given as mean ± SD or median [IQR] depending on the distribution. Categorical measures are shown as frequencies (percentages).

### 3.2. CMV IgG Levels across the Study Arms

CMV IgG differed significantly across study groups (Figure 1, *p* = 0.003). Considering pairwise comparisons (using the Bonferroni adjustment for multiple comparisons), the mean CMV IgG among the ART-naïve group was significantly higher than both the on-cART group (*p* = 0.02) and the controls (*p* = 0.002). The mean CMV IgG did not differ significantly between the controls and on-cART groups (*p* = 1.0).

### 3.3. The Association between CMV IgG and Demographic and Clinical Characteristics

The associations between CMV IgG and certain covariates of interest (age, CD4, ART, gender, waist–hip ratio, smoking, hypertension, HbA1c) were individually assessed overall and separately for each study group. There were no significant associations with CMV IgG (Appendix A).

### 3.4. The Association between CMV IgG Levels and Systemic Inflammation/Arterial Disease (Endothelial Injury) by Study Groups

In multivariable models for arterial disease, the interaction between CMV IgG and ART-naïve PLWH was significantly associated with increases in log VCAM-1 (*p* = 0.01); however, there was no association between log VCAM-1 and CMV IgG among the other two groups (Table 2). CMV IgG levels were associated with ICAM-1 levels among ART-naïve PLWH (*p* = 0.01) and PLWH on ART (*p* < 0.01), relative to HIV-uninfected controls. CMV IgG levels were associated with cIMT among PLWH on ART (*p* = 0.03), relative to the HIV-uninfected controls. The rest of the results are provided in Table 2 below.

## 4. Discussion

This study investigated the association between CMV IgG levels and three distinct HIV status groups in Botswana. Additionally, we investigated the associations between CMV IgG levels and carotid intimal media thickness (cIMT), as well as biomarkers of endothelial injury and monocyte activation, among the three groups.

Consistent with previous studies [6,7,11,22,23,24], we found that CMV IgG levels are significantly associated with ART naivety among PLWH. cIMT was not associated with any of the three HIV sub-groups before considering the distribution of CMV IgG levels. However, after considering the distribution of CMV IgG levels across the three study groups, we observed a signal that CMV IgG levels were correlated with cIMT among PLWH on ART, relative to the HIV-uninfected group in this setting. CMV IgG levels were also associated with VCAM-1 and ICAM-1, but not E-selectin, among the ART-naïve PLWH. Additionally, CMV IgG levels were not associated with sCD163, a marker of monocyte activation, among PLWH relative to the controls.

Our study found CMV IgG levels to be associated with cIMT (measured in the CCA) among PLWH on ART relative to the HIV-uninfected controls. This finding aligns with some previous studies [16,17], but with some heterogeneity in their comparison groups. For instance, in the study by Hsue et al. [16], they examined 93 PLWH, of whom only 57% had undetectable HIV VL. CMV seroprevalence was 99% among PLWH and 92% among HIV-uninfected controls in their study. On the other hand, Knudsen et al. [17] reported CMV seroprevalence rates of 57% among PLWH and 68% among HIV-uninfected controls. In contrast, all our participants had CMV seroprevalence, and 100% of those on ART had undetectable HIV viral loads.

Other studies assessing cIMT among PLWH did not assess CMV co-infection and are therefore not comparable with our study. Nevertheless, these studies have produced conflicting results, with some showing higher IMT among PLWH [25,26,27], while others negated this observation [28,29,30,31], in part due to wide heterogeneity in the methodology, including the level of cIMT measurement (CCA vs. bifurcation). These varying findings highlight the complexity of the relationship between HIV infection, antiretroviral therapy, and cIMT in different arterial segments.

It is unclear why we found a significant association between higher cIMT among PLWH on ART relative to the controls but not among PLWH who were ART-naïve. One possible explanation is that, ART, particularly protease inhibitors, alter the lipid profile and the subsequent development of atherosclerosis [32,33]. Although we did not assess the temporality between the lipid profile in our study and ART, we observed higher lipid profile results among participants on ART than the other groups.

In our study, PLWH with CMV seroprevalence who either received ART or were ART-naïve showed higher ICAM-1 and or log VCAM-1 levels, biomarkers of endothelial injury, than their age-, gender-, smoking-, and anthropometry-matched controls with CMV seroprevalence. Many other studies on endothelial injury in Africa [31,34,35] and beyond [36] did not assess the relationship between endothelial injury and CMV IgG. This highlights the novelty of our study in providing a possible explanatory variable between CMV seroprevalence among PLWH and vascular inflammation or endothelial injury. Assessing vascular inflammation or injury among PLWH and its associated risk factors has become increasingly important, particularly given recent strategies aimed at cardiovascular risk in PLWH through the use of statins [37].

Our study’s finding that there was no association between CMV-HIV co-infection and sCD163 in either the ART-naïve or on ART-treated participants differs from a US (Chicago) study of three cohorts with CMV IgG: 54 HIV-uninfected people, 58 PLWH but aviremic (94% on ART), and 52 PLWH and viremic. The study found higher sCD163 levels, *p* < 0.001, among the HIV-1 viremic group than either the aviremic or uninfected group [23]. Our findings also differ from a Boston, USA, study of CMV IgG-positive 41 HIV-uninfected controls and 102 PLWH (including 69 on ART with undetectable VL) which showed higher mean levels of sCD163 among the PLWH than controls, *p* = 0.006 [22]. Key differences between their study and ours include a predominantly white population (>60% vs. 0%) and an overall higher proportion of participants with family history of premature coronary heart disease (13–22% vs. 0–5%), hypertension (12–26% vs. 0–15%), current smoking (29–41% vs. 9–20%), and long HIV disease duration (13.8 years vs. 9.8 years) but less ART duration (7.2 years vs. 8.6 years) and protease inhibitor use (52% vs. 255) [22].

### Limitations and Strengths

Our study has notable limitations, including its cross-sectional design, which restricts causal interpretations, and the small sample size, limiting generalizability. The study population was relatively young since individuals aged >50 were not included. Additionally, the cross-sectional nature of this study means that there was no assessment on the temporal relationship between ART use and dynamic biomarkers such as lipid profile, markers of endothelial injury, and cIMT. Focusing on cIMT at the common carotid artery alone may also have led to missing early atherosclerosis, as it typically manifests later at this site and is likely to be less common in younger participants [38]. This is an observational study, and it is possible that associations of CMV or HIV infection with markers of subclinical atherosclerosis may not accurately represent causal relationships. Finally, the evaluation of additional cytokines and biomarkers that likely play a role in chronic inflammation in HIV infection would provide a broader assessment of the inflammatory milieu and would be of interest.

Nonetheless, this study among an established HIV cohort has several strengths, including extensive data on potential confounders and metabolic risk mediators and the inclusion of an HIV-uninfected comparison group recruited using comparable methods and from similar venues as the HIV-infected individuals. Furthermore, our study enrolled patients with a relatively long duration of largely well controlled HIV infection, with minimal traditional risks and without known carotid disease. 

## Figures and Tables

**Figure 1 biomedicines-12-00935-f001:**
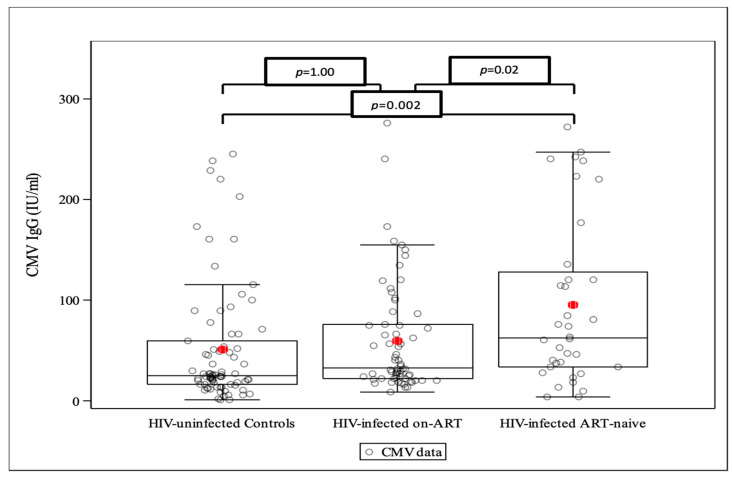
CMV IgG levels across study arms (the horizontal bars of the whiskers represent the 10th and 90th percentile, while the upper and lower boundaries of the boxes represent the 1st and 3rd quartiles. The dots on either side of the box-plot represent the lower and upper extremes of the data). The red dots represents the mean CMV IgG levels.

**Table 1 biomedicines-12-00935-t001:** Demographic and clinical characteristics of participants.

	HIV-Uninfected Controls, *n* = 79	PLWH on-cART, *n* = 68	PLWH Who Were ART-Naïve, *n* = 36	*p*-Value
N	79	68	36	0.653
Age				
Mean (SD)	38.2 (5.4)	38.7 (4.6)	37.8 (5.4)	
30–35	32 (41.0%)	16 (24.0%)	16 (44.0%)	0.027
36–40	21 (27.0%)	34 (50.0%)	10 (28.0%)	
41–50	26 (33.0%)	18 (26.0%)	10 (28.0%)	
Female Sex, N (%)	36 (45.6%)	35 (51.5%)	21 (58.3%)	0.434
Cigarette Smoking, N (%)				
Ever	27 (34.0%)	27 (40.0%)	7 (19.0%)	0.102
Current	16 (20.0%)	6 (9.0%)	4 (11.0%)	0.134
Family history, N, (%)				
CAD	4( 5.0%)	1 (1.0%)	0	0.278
Stroke	16 (20.0%)	6 (9.0%)	1 (3.0%)	0.020
Anthropometric data, mean (SD)				
SBP (mmHg)	130.5 (17.3)	128.5 (16.3)	129.8 (17.5)	0.768
DBP (mmHg)	84.7 (13.5)	83.9 (12.2)	85.2 (13.5)	0.877
Waist–Hip ratio, N(%)				
Male (≥0.90)	7 (16.0%)	8 (24.0%)	7 (47.0%)	0.065
Female (≥0.85)	14 (39.0%)	17 (50.0%)	6 (29.0%)	0.300
Medical History, N (%)				
Diabetes mellitus	0	1 (1.0%)	1 (3.0%)	0.322
Hypertension	10 (13.0%)	10 (15.0%)	0	0.031
Chronic kidney disease	0	2 (3.0%)	0	0.175
Dyslipidemia	2 (3.0%)	6 (9.0%)	0	0.093
CVD Risk Blood Tests, mean (SD)				
Total cholesterol (mmol/L)	4.4 (1.2)	4.8 (1.3)	4.1 (0.9)	0.003
LDL cholesterol (mmol/L)	2.5 (1.0)	2.9 (1.1)	2.3 (0.8)	0.020
HDL cholesterol (mmol/L)	1.4 (0.4)	1.5 (0.5)	1.2 (0.4)	0.003
Non-HDL cholesterol (mmol/L)	2.9 (1.2)	3.3 (1.3)	2.9 (0.9)	0.059
Triglycerides (mmol/L)	1.2 (1.5)	1.4 (0.8)	1.2 (0.6)	0.541
HbA1c (%) (SD)	5.5 (1.0)	5.3 (0.4)	5.7 (0.5)	0.021
HIV Parameters, mean (SD)				
HIV disease duration (yrs)	N/A	9.8 (3.2)	N/A	N/A
ART duration (yrs)	N/A	8.6 (2.8)	N/A	N/A
CD4 nadir (cells/uL)	N/A	123.5 (76)	N/A	N/A
Baseline CD4 count (cells/uL)	N/A	122.9( 73.1)	N/A	
Current CD4 count (cells/uL)	N/A	540.2 (230.8)	381.5 (236.6)	0.001
Proportion with undetectable VL	N/A	100%	N/A	N/A
Months since VL < 400 cp/mL	N/A	3.4 (2)	N/A	N/A
Patients on NNRTI-based ART	N/A	50 (75.0%)	N/A	N/A
Patients on PI-based ART	N/A	17 (25.0%)	N/A	N/A
Carotid Assessment, mean (SD)				
Mean cIMT (mm) (SD)	0.616 (0.1)	0.595 (0.09)	0.574 (0.06)	0.059
Any plaque, N (%)	2 (3.0%)	0	2 (6.0%)	0.190
Biomarkers, mean (SD)				
Soluble CD163 (ng/mL)	7.1 (0.6)	7.0 (0.7)	7.3 (0.5)	0.069
ICAM-1 (ng/mL)	446.3 (164.2)	529.1 (156.6)	574.1 (203.9)	<0.001
Log-VCAM-1 (log ng/mL)	6.5 (0.4)	6.6 (0.3)	7.0 (0.5)	<0.001
e-Selectin (ng/mL)	63.8 (18.4)	61.6 (23.4)	58.4 (32.5)	0.710

HIV: human immunodeficiency virus, SBP: Systolic Blood Pressure, DBP: Diastolic Blood Pressure, PLWH: people living with HIV, LDL: low-density lipoprotein, HDL: high-density lipoprotein, PI: protease inhibitor, NNRTI: non-nucleoside reverse transcriptase inhibitor, CAD: Coronary Artery Disease, ART: anti-retroviral therapy, cIMT: carotid intimal media thickness, soluble CD163: soluble cluster of differentiation 163, VCAM-1: vascular adhesion molecule, ICAM-1: intercellular adhesion molecule 1.

**Table 2 biomedicines-12-00935-t002:** The association between CMV IgG and arterial disease by study group.

		log VCAM-1			ICAM-1		e-Selectin		cIMT		sCD163	
Model		Est (95% CI)	*p*		Est (95% CI)	*p*	Est (95% CI)	*p*	Est (95% CI)	*p*	Est (95% CI)	*p*
1	CMV IgG	0.001 (0, 0.002)	0.2		0.2 (−0.3, 0.6)	0.4	−0.01 (−0.08, 0.06)	0.8	0 (0.0)	0.09	0 (−0.001, 0.002)	0.8
2 *				CMV IgG	0.09 (−0.4, 0.5)	0.7	0.009 (−0.07, 0.08)	0.8	0 (0.0)	0.2	0 (−0.001, 0.002)	0.8
	CMV among PLWH on ART	−0.001 (−0.002, 0.001)	0.4	PLWH on-ART	88.4 (26.6, 150.3)	<0.01	1.07 (−8.5, 10.7)	0.8	−0.03 (−0.06, −0.003)	0.03	−0.2 (−0.4, 0.02)	0.1
	CMV among ART-naïve PLWH	0.002 (0, 0.004)	0.01	ART-naïve PLWH	102.2 (24.2, 180.2)	0.01	−12.1 (−27.4, 3.2)	0.1	−0.03 (−0.06, 0.004)	0.09	0.03 (−0.2, 0.3)	0.9
	CMV among controls	0 (−0.002, 0.001)	0.5	HIV-uninfected Controls	(ref)		(ref)		(ref)		(ref)	

* Additionally adjusted for age, gender, currently smoking cigarettes, waist–hip ratio, glycosylated hemoglobin, participant history of hypertension, and non-HDL cholesterol. Abbreviations: HIV: human immunodeficiency virus; PLWH: people living with HIV; LDL: low-density lipoprotein; ART: anti-retroviral therapy; cIMT: carotid intimal media thickness; sCD163: soluble cluster of differentiation 163; VCAM-1: vascular adhesion molecule; ICAM-1: intercellular adhesion molecule 1.

## Data Availability

Data access is restricted due to ethical restrictions imposed by the Ministry of Health Ethics Committee. Study data are available through an application to the Botswana Ministry of Health Human Research Development Committee (HRDC). Contacts for the Committee: Email: hhealthresearch@govbots.onmicrosoft.com. Telephone: +267-391-4467/363-2751.

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
