# Peer review of "Cytomegalovirus Immunoglobulin G Levels and Subclinical Arterial Disease among People Living with HIV in Botswana: A Cross-Sectional Study"

_biomedicines, 2024, doi:10.3390/biomedicines12050935_

Round 1
Reviewer 1 Report
Comments and Suggestions for Authors
Cytomegalovirus Immunoglobulin G levels and subclinical arterial disease among people living with HIV in Botswana: a cross-sectional study
Manuscript ID: biomedicines-2883794
The authors evaluated and compared the CMV immunoglobulin G (IgG) levels between ART-treated PLWH versus ART-naïve PLWH and 16 those without HIV, and to investigated their associations with biomarkers of endothelial injury and carotid atherosclerosis, in Botswana. The results demonstrated that the mean CMV IgG level among ART-naïve participants was significantly higher than both the cART group and controls. However, CMV IgG levels did not differ significantly between the controls and cART groups.
The authors have described the study findings extremely well. The introduction section, results, and the conclusion section are well described. No further corrections are required.
Comments on the Quality of English LanguageMinor editing of English language required
Author Response
Comments: The authors evaluated and compared the CMV immunoglobulin G (IgG) levels between ART-treated PLWH versus ART-naïve PLWH and those without HIV, and investigated their associations with biomarkers of endothelial injury and carotid atherosclerosis, in Botswana. The results demonstrated that the mean CMV IgG level among ART-naïve participants was significantly higher than both the cART group and controls. However, CMV IgG levels did not differ significantly between the controls and cART groups. The authors have described the study findings extremely well. The introduction section, results, and the conclusion section are well described. No further corrections are required.
Response: Thank you very much for your comments. We are glad you found it insightful.
Reviewer 2 Report
Comments and Suggestions for Authors
I truly enjoyed reading this exploratory analysis of previously acquired dataset, investigating CMV serology in people living with HIV and associated subclinical arterial disease. The study is overall well performed, transparently presented. Limitations are adequately recognized and the results are interpreted critically. I suggest only minor improvements. Table formatting needs to be revisited since they are hard to read and misaligned. Also, there seem to be some errors in table 1 presentation since extremely high number for age are presented in column regarding control patients, however, this my be due to table formatting. Authors could also comment on recent strategies aimed at reducing cardiovascular risk in people living with HIV, considering recent randomized data that suggest benefit with statin use (DOI: 10.1056/NEJMoa2304146)
Author Response
Comment 1: I truly enjoyed reading this exploratory analysis of previously acquired dataset, investigating CMV serology in people living with HIV and associated subclinical arterial disease. The study is overall well performed, transparently presented. Limitations are adequately recognized, and the results are interpreted critically.
Response: We are glad you enjoyed reading this paper. We hope many readers will enjoy it too.
Comment 2: I suggest only minor improvements. Table formatting needs to be revisited since they are hard to read and misaligned. Also, there seem to be some errors in table 1 presentation since extremely high number for age are presented in column regarding control patients, however, this my be due to table formatting.
Response: Thank you very much for pointing out these errors that occurred during table formatting. The errors have now been fixed and the table has been improved, see Table 1 (attached) below and in the revised manuscript.
Comment 3: Authors could also comment on recent strategies aimed at reducing cardiovascular risk in people living with HIV, considering recent randomized data that suggest benefit with statin use (DOI: 10.1056/NEJMoa2304146)
Response: Thank you for this valuable suggestion. We now include the statement “Assessing vascular inflammation or injury among PLWH and its associated risk factors has become increasingly important, particularly given recent strategies aimed at reducing cardiovascular risk among PLWH through the use of statins”, in the discussion section, page 9, line 306-309, ref “Grinspoon SK, Fitch KV, Zanni MV, Fichtenbaum CJ, Umbleja T, Aberg JA, et al. Pitavastatin to Prevent Cardiovascular Disease in HIV Infection. New England Journal of Medicine. 2023;389(8):687-99.”

Reviewer 3 Report
Comments and Suggestions for Authors
The authors provide a selected cohort of PLWH divided according to treatment with ART or not and controls from Botswana. i presume they are ethnically homogeneous. They are imbalanced in numbers due I presume to whether they had serum available for CMV estimates.
They claim that higher CMV IgG levels associate with some measures of endothelial function but not others, and that cIMT is associated with this CMV in some groups but not all.
The findings are not critically novel and limitations of such a study are great. My major concern is that the authors neglect to consider how the finding that cIMT is not increased in the PLWH groups compared to the controls, if anything it is lower (p=0.059 table 1) and in fact the entire population cIMT is very low perhaps reflecting the young age of the patients involved. A high risk group is often considered >0.9 for example although whether this is known in Africans is uncertain. On this basis, the major determinant of "vascular risk" is NOT increased but diminished. The authors do not highlight this fact but continue to suggest that there is increased cIMT in some groups and link this with CMV by using within group comparisons.
I suggest that constructing a regression using cIMT as the outcome for the entire group and considreing risks factors for this (age, sex, lipids, BP etc) including PLWH group assignment looking for interactions is more appropriate. By a priori determining to separate groups and provide within group analyses of multiple risks one will, by chance, likely finding a positive correlation.
I suggest the authors acknowledge that there is no difference in cIMT between the groups and therefore modify some of their claims as to its relevance to associations with vascular risk.
Its crucial to avoid drawing any hint of causality between associations in such studies (examples listed below)
Further points:
p3 lines 128-129- why are the 5 outliers removed?
p3 lines 140/141- Please correct and remove "the impact of CMV to "the association of CMV"
p4 lines 164-166- again the reasons for deleting these cases isnt made clear. Can that authors state in addition whether they detected patients who had no detectable CMV IgG (number and by group assignemtn) and how these were handled in the analyses?
Figure 1 - would the authors consider presenting all of the data including cIMT and the VCAM/ICAM and 'seltecins in a similar manner to the CMV data so the readers can better view the distributions please? I think the authors allow the cIMT data to be hidden in the table- I prefer all data presented so the reader can analyse the distributions.
The multivariable modelling- I think this needs better explanations- it appears each group has its own MV model but what is included, what is adjusted for and what is significant is not clear to me. Plesae explain this with greater precision. please justify why the data is analysed by group and not as a whole?
p7 liness 223-224- This sentence is better expressed ass "CMV IgG levels correlated with with cIMT in PLWH on ART" the authors use of higher and increasing implies that they are abnormal- there is simply a correlation- there is no causation here and the authors have not dealt with the issue that cIMT is lower in the PLWH groups than the controls- the impression is that the PLWH have worse cIMT and higher risk- this is not correct.
p7 Lines 229-230- again the authors claims here are not supported by the data- they state that the higher CMV is associated with higher cIMT when this is not correct- they show an association within 1 group not seen in the other but overall the cIMT is not increased. Please adjust. Please discuss this more objectively and comment on why this migth be the case.
Comments on the Quality of English LanguageThe english has only minor flaws.
Author Response
Comment 1: The authors provide a selected cohort of PLWH divided according to treatment with ART or not and controls from Botswana. i presume they are ethnically homogeneous. They are imbalanced in numbers due I presume to whether they had serum available for CMV estimates.
Responses:
It is true that Botswana’s population is largely black race and ethnically homogenous. We have now included this information in study participants section, page 2, lines 68-69.
Indeed, the imbalance in the number of participants per group was due to the availability of CMV Immunoglobulin per HIV sub-group from the source study population groups, now included in page 4, lines 178-181. Additionally, the low numbers of participants who were ART naïve reflect our nation-wide high uptake of ART (having attained UNAIDS 95:95:95% targets), page 2, line 77-79.
Comment 2: They claim that higher CMV IgG levels associate with some measures of endothelial function but not others, and that cIMT is associated with this CMV in some groups but not all. The findings are not critically novel and limitations of such a study are great. My major concern is that the authors neglect to consider how the finding that cIMT is not increased in the PLWH groups compared to the controls, if anything it is lower (p=0.059 table 1) and in fact the entire population cIMT is very low perhaps reflecting the young age of the patients involved. A high risk group is often considered >0.9 for example although whether this is known in Africans is uncertain. On this basis, the major determinant of "vascular risk" is NOT increased but diminished. The authors do not highlight this fact but continue to suggest that there is increased cIMT in some groups and link this with CMV by using within group comparisons. I suggest that constructing a regression using cIMT as the outcome for the entire group and considreing risks factors for this (age, sex, lipids, BP etc) including PLWH group assignment looking for interactions is more appropriate. By a priori determining to separate groups and provide within group analyses of multiple risks one will, by chance, likely finding a positive correlation.
Response:
Thanks for this comment. We acknowledge that in our study cIMT was not associated with any of the three HIV sub-groups (p=0.059) before considering the distribution of CMV IgG levels (our exposure of interest). However, after considering the distribution of CMV IgG levels across the three study groups, we observed a signal that there was a statistically significant association between CMV IgG among PLWH on ART with cIMT relative to the HIV uninfected but CMV IgG positive group. This statement is now included and clarified in the revised manuscript, page 8, lines 254-258.
Furthermore, we would like to clarify that this study was not powered to study an association between cIMT and HIV but rather to examine an association between CMV IgG and markers of endothelial injury, including cIMT, across two distinct groups of HIV positive patients and a control group. The main study from which this current study is embedded is the one that was powered and aimed to assess the relationship between cIMT and HIV, and indeed showed that there was no significant difference in cIMT between HIV infected participants and HIV uninfected controls (DOI: https://doi.org/10.1371%2Fjournal.pone.0179994).
Specific to this exploratory study, we make inferences about level of CVD risk based on relative differences between our study measures- this inferences are restricted to the current study groups and discuss “higher risk” and not absolute “high risk based on cIMT cut off of 0.9 from other non-African cohorts” because we are not aware of cIMT cut off points for those at risk of developing major cardiovascular end points (MACE) in this setting.
Comment 3: I suggest the authors acknowledge that there is no difference in cIMT between the groups and therefore modify some of their claims as to its relevance to associations with vascular risk. It’s crucial to avoid drawing any hint of causality between associations in such studies (examples listed below).
Response: Thank you, we have included this statement in the revised manuscript, page 4, line 183-184.
Comment 4: p3 lines 128-129- why are the 5 outliers removed?
Response: These outliers were excluded at the request of the laboratory as they were concerned for a laboratory error affecting the 5 specimens, now clarified on page 3, line 134.
Comment 5: p3 lines 140/141- Please correct and remove "the impact of CMV to "the association of CMV"
Response: Thank you, this has been modified, page 3, line 144.
Comment 6: p4 lines 164-166- again the reasons for deleting these cases isnt made clear. Can that authors state in addition whether they detected patients who had no detectable CMV IgG (number and by group assignemtn) and how these were handled in the analyses?
Response: All the participants we tested had detectable CMV IgG.
Comment 7: Figure 1 - would the authors consider presenting all of the data including cIMT and the VCAM/ICAM and 'seltecins in a similar manner to the CMV data so the readers can better view the distributions please? I think the authors allow the cIMT data to be hidden in the table- I prefer all data presented so the reader can analyse the distributions.
Response: As stated above, our main aims were to compare CMV IgG levels (exposure of interest) among PLWH and a matched control group and to assess the associations between CMV IgG with markers of endothelial injury, cIMT and monocyte activation (outcomes of interest), as such figure 1 and table 2 are meant to reflect those associations. Please note that reference to detailed cIMT data from associated work, which was appropriately powered to assess differences in cIMT by group, was included as reference no. 19 (https://doi.org/10.1371%2Fjournal.pone.0179994) in the manuscript.
Comment 8: The multivariable modelling- I think this needs better explanations- it appears each group has its own MV model but what is included, what is adjusted for and what is significant is not clear to me. Plesae explain this with greater precision.
- What is included; Response: The multivariable models for each outcome included CMV IgG, study group, and any variables with a p-value<0.15 or deemed to be confounders. Each of the final multivariable models was re-assessed among the on ART experienced group only and separately adjusted for ART duration, number of ART switches, type of ART regimen, and nadir CD4. Please see our statistical analysis section on page 4, lines 163-167 in the manuscript where these details are included.
- What is adjusted for; Response: our multivariable models assessed for potential confounders, including age, sex, ART duration, number of ART switches, PI- versus NNRTI-containing regimen, nadir CD4, current CD4, waist-hip ratio, cigarette smoking, diagnosis of hypertension and HBA1C. Please see page 4, line 135-139 in the statistical analysis section of the manuscript where these details were previously presented.
- What is significant; Response: CMV IgG levels were associated with log VCAM-1 (p-value=0.01) and ICAM-1 (p=0.01), but not E-selectin (p=0.1), sCD163 (p=0.9) and cIMT (p=0.09), among the ART-naïve PLWH relative to the controls. CMV IgG levels were also associated with ICAM-1 (p=0.01) and cIMT (p=0.03) among ART PLWH on-ART relative to the controls. Please see page 7-8, lines 232-240, and table 2, for this detailed information, as was previously presented in the manuscript.
Comment 9: please justify why the data is analysed by group and not as a whole?
Response: The data is analysed per group and not as a whole because of our aims; to compare CMV IgG levels among three groups of participants, PLWH on ART, PLWH-ART naive and a matched control group, and to assess the association between CMV IgG with markers of endothelial injury, cIMT and monocyte activation. Assessing the group as a whole would not answer our aims. Please note that these three groups are potentially very different immunologically and assessing them as a whole may not be consistent with basic biologically plausible assumptions about immunologic status of HIV-uninfected versus long term ART treated versus ART naïve people living with HIV.
Comment 10: p7 liness 223-224- This sentence is better expressed ass "CMV IgG levels correlated with with cIMT in PLWH on ART" the authors use of higher and increasing implies that they are abnormal- there is simply a correlation- there is no causation here and the authors have not dealt with the issue that cIMT is lower in the PLWH groups than the controls- the impression is that the PLWH have worse cIMT and higher risk- this is not correct.
Response: Thank you for this suggestion, the sentence has been improved accordingly. It reads, “cIMT was not associated with any of the three HIV sub-groups before considering the distribution of CMV IgG levels. However, after considering the distribution of CMV IgG levels across the three study groups, we observed a signal that CMV IgG levels were correlated with cIMT among PLWH on ART, relative to the HIV uninfected group in this setting.”, See page 8, line 254-258.
Comment 11: p7 Lines 229-230- again the authors claims here are not supported by the data- they state that the higher CMV is associated with higher cIMT when this is not correct- they show an association within 1 group not seen in the other but overall the cIMT is not increased. Please adjust. Please discuss this more objectively and comment on why this migth be the case
Response: Thank you for this observation. The sentence has been adjusted accordingly to read “Our study found CMV IgG levels to be associated with cIMT (measured in the CCA) among PLWH on ART relative to the HIV-uninfected controls ”. See page 8, line 262-263.
Reviewer 4 Report
Comments and Suggestions for Authors
This retrospective cross-sectional study compares CMV IgG levels in a well-characterized cohort of adult people living with HIV (PLWH) and a matched group of HIV uninfected individuals in Gaborone, Botswana, and to investigate their associations with biomarkers of endothelial injury and carotid atherosclerosis. Although the study design has several limitations that are recognized by the authors (last paragraph of Discussion), it provides valuable aspects that deserve to be highlighted.
Author Response
Thank you very much for taking your time to review this manuscript. We have made revisions to the methods, results and conclusions accordingly. Please also find the responses to Reviewer 3 (above) and the corresponding revisions/corrections highlighted in the re-submitted files.